# Synthesis of Novel Arginine-Based Flame Retardant and Its Application in Lyocell Fabric

**DOI:** 10.3390/molecules26123588

**Published:** 2021-06-11

**Authors:** Jiayi Chen, Yansong Liu, Jiayue Zhang, Yuanlin Ren, Xiaohui Liu

**Affiliations:** 1School of Textiles Science and Engineering, Tiangong University, Tianjin 300387, China; chenjiayi20210506@163.com (J.C.); yansongliu1996@163.com (Y.L.); keyyoume@163.com (J.Z.); 2Key Laboratory of Advanced Textile Composite, Ministry of Education, Tiangong University, Tianjin 300387, China; 3School of Materials Science and Engineering, Tiangong University, Tianjin 300387, China; xiaohuilau@163.com

**Keywords:** lyocell fabric, arginine, flame retardant, thermal properties, mechanism

## Abstract

Lyocell fabrics are widely applied in textiles, however, its high flammability increases the risk of fire. Therefore, to resolve the issue, a novel biomass-based flame retardant with phosphorus and nitrogen elements was designed and synthesized by the reaction of arginine with phosphoric acid and urea. It was then grafted onto the lyocell fabric by a dip-dry-cure technique to prepare durable flame-retardant lyocell fabric (FR-lyocell). X-ray photoelectron spectroscopy (XPS) and Fourier-transform infrared spectroscopy (FTIR) analysis demonstrated that the flame retardant was successfully introduced into the lyocell sample. Thermogravimetric (TG) and Raman analyses confirmed that the modified lyocell fabric featured excellent thermal stability and significantly increased char residue. Vertical combustion results indicated that FR-lyocell before and after washing formed a complete and dense char layer. Thermogravimetric Fourier-transform infrared (TG-FTIR) analysis suggested that incombustible substances (such as H_2_O and CO_2_) were produced and played a significant fire retarding role in the gas phase. The cone calorimeter test corroborated that the peak of heat release rate (PHRR) and total heat release (THR) declined by 89.4% and 56.4%, respectively. These results indicated that the flame retardancy of the lyocell fabric was observably ameliorated.

## 1. Introduction

Lyocell fibers have been known as the green fibers of the 21st century. For the issue of environmental protection, wet-spun cellulose fibers derived from green solvent were first studied in the early1980s [1]. Lyocell fibers are fabricated by first dissolving cellulose in N-methylmorpholine-N-oxide/aqueous solution to form an even spinning solution, then it was extruded through a spinneret, and immersed into a coagulation bath to form lyocell fibers by drawing [2]. Compared with the traditional viscose process, the production of lyocell fibers is a simple and resource-saving method for the production of regenerated cellulose fibers [3]. Lyocell fibers belong to regenerated cellulose fibers and can be completely degraded due to their specific structure [4,5,6]. Lyocell fibers possess a number of features, such as the advantages of both natural and synthetic fibers, as well as the comfort of cotton, the drape and bright color of viscose, the soft handle and elegant gloss of silk. Therefore, the lyocell fibers are extensively applied in the textile field [7,8]. Lyocell fibers, however, are a rich source of hydrocarbons, therefore, they are flammable and prone to cracking at high temperature, causing serious fire hazards and incalculable losses, which greatly limits their wide application [9,10,11,12,13].

Accordingly, considering the safety and practicality, the development of cellulose fibers and/or fabrics with good flame retardancy is of great significance [14,15]. However, it is found that most flame retardant finishing agents have adverse effects on the environment and human health because of the contained carcinogenic chemical groups [16]. Halogenated flame retardants can effectively capture high energy free radicals during thermal decomposition and play the role of flame retardancy [17]. Nevertheless, many halogen-based flame retardants have been or are being phased out because highly toxic substances are generated during burning, causing severe environmental contamination and health concerns [18,19]. Nevertheless, in the combustion process, boron-based flame retardants can produce gas–liquid intermediates to form an impermeable layer, acting as a barrier of heat and oxygen. However, the fire resistance and durableness of cellulose fabrics treated with boron-based flame retardant are not satisfactory [20]. Silicone-containing flame retardants form an amorphous silicon protective layer on the surface of the material after burning to isolate the transfer of air and oxygen. Amorphous silica dioxide (SiO_2_), due to its high heat resistance, inert and non-toxic properties, is considered as a potential and safe flame retardant [21]. The combination of silicon with other flame retardants will play a synergistic flame-retardant effect. Jiang et al. used sol–gel technology to construct an expansion flame retardant coating containing phosphorus, nitrogen, and silicon elements, which has a good flame-retardant effect on PAN fabric [22]. Metal hydroxides, particularly magnesium hydroxide and aluminum hydroxide, are used as environmentally friendly flame retardants to reduce the temperature of materials by endothermic reaction without producing toxic gases. They have the advantages of abundant resources, cost effectiveness, super heat endurance and smoke suppression performance [23,24]. Nevertheless, metal hydroxides are not highly suitable flame retardants for cellulose fibers or fabrics [25]. On the contrary, phosphorus-based flame retardants possess some superiorities, for example, high efficiency, low toxicity, low cost and being environmentally friendly [26,27]. Moreover, phosphorus-containing flame retardants can effectively minify the pyrogenic decomposition of cellulose and promote the dehydration and carbonization of cellulose. The char layer screens oxygen and heat, effectively inhibiting or interrupting the combustion process [28]. So far, flame retardants containing phosphorus and/or nitrogen have been identified as a substitute for halogen flame retardants and are widely used in commercial applications [13,29].

Recently, many biomass compounds containing nitrogen and/or phosphorus have been assembled into environmentally friendly flame retardants, for instance, date stones, eggs, rice husks, DNA, banana pseudostem liquid and spinach juice, and so on [30,31,32,33]. Renewable biomass resources, such as proteins, with the advantages of green, renewable and sustainable, are ideal substitutes for halogenated flame retardants [34]. Amino acids are the basic structural units of proteins and are an indispensable part of the human body. Together with nucleic acids, carbohydrates and lipids, they play an extremely important role in human physiological processes. Arginine is a typical green renewable resource, mainly derived from grain. Based on the reactive functional groups of amino group and carboxyl group, arginine has high biological activity and is widely applied in the fields of medicine, food, agriculture, cosmetics, and so on [32]. Therefore, in this work, arginine and urea were used as nitrogen sources, and phosphoric acid with high phosphorus content was used as an acid source to successfully fabricate a biomass-based flame retardant for lyocell fabrics. Then the modified lyocell fabric obtained high efficiency and durable flame retardancy. The structure, thermal properties, and combustion properties were all studied in detail.

## 2. Materials and Methods

### 2.1. Materials

Lyocell fabric with the basic weight of 150 g/m^2^ was fabricated by textile laboratory of Tiangong University (Tianjin, China). Urea and phosphoric acid (85%) were provided by Tianjin Fengchuan Chemical Reagent Co. Ltd. (Tianjin, China). Dicyandiamide was acquired from Tianjin Guangfu Fine Chemical Research Institute (Tianjin, China). Anhydrous ethanol and arginine were supplied by Tianjin Kermal Chemical Reagent Co., Ltd. (Tianjin, China). All the reagents were analytical grade and used without any treatment.

### 2.2. Synthesis of Flame Retardant

Arginine (0.04 mol, 6.968 g) and 30 mL deionized water were added in a three-necked round bottom flask with a magnetic stirring device and a spherical reflux condenser. Then, phosphoric acid (0.08 mol, 7.84 g) was put into the flask, heated to 100 °C in an oil bath and it reacted for 3 h under stirring. After that, urea (0.2 mol, 12.012 g) was slowly added into the reaction system and maintained at 100 °C for 2 h. Then a transparent yellow liquid was cooled down to room temperature, rinsed three times with anhydrous ethanol. Finally, a white solid flame retardant was obtained after extraction and vacuum drying at 60 °C, i.e., the flame retardant was obtained. The theoretical yield of flame retardant was 18.08 g, and the actual yield of the three repeated tests was 14.60, 15.03, and 15.12 g, respectively, with an average yield of 82.50%. The synthetic reaction of flame retardants is presented in Figure 1.

### 2.3. Manufacture of Flame-Retardant Lyocell Fabric (FR-Lyocell)

The preparation scheme of flame-retardant lyocell fabric was illustrated in Figure 2. First, the flame retardant was dissolved in distilled water to obtain the flame retardant solutions with the concentration of 120 g/L, 150 g/L, and 180 g/L, respectively. The lyocell fabric treated with 180 g/L flame retardant solution was mainly studied. Secondly 7 wt% of dicyandiamide was placed into the solution as acatalyst, then the mixed solution was put into a thermostatic water bath at 80 °C and stirred until the dicyandiamide was dissolved completely. Afterwards, the lyocell fabric was immersed in the solution at a bath ratio of fabric mass to graft liquid volume, i.e., 1:50 at 80 °C for 3 h. Next, the lyocell fabric was taken out and baked in an oven at 170 °C for 10 min. Subsequently, the unreacted flame retardant was washed away with deionized water. Finally, the treated lyocell sample was dried to a constant weight in an oven at 60 °C. The weight gain (WG) of the FR-lyocell fabric was calculated using the following formula:WG (%) = [(m_2_ − m_1_)/m_1_] × 100%(1)
where m_1_ and m_2_ are the weights of lyocell fabric before and after flame-retardant treatment, respectively. The weight gain (WG) is listed in Table 1.

### 2.4. Characterizations

X-ray photoelectron spectroscopy was utilized to explore the surface elemental contents of all the specimens by using an AXIS-Ultra DLD XPS spectrometer (K-alpha, Waltham, MA, USA).

The surface morphology of all the specimens was probed by field emission scanning electron (FE-SEM, S-4800, Hitachi Ltd., Tokyo, Japan), fitted with energy-dispersion spectrometer (EDS) for element mapping analysis. Scanning electron microscopy (SEM) images of the specimens were observed using a S4800 field-emission SEM at 10 kV accelerating voltage (Hitachi Co, Japan).

Fourier-transform infrared spectroscopy (JASCO, FTIR-6300, spectrometer, UK) was employed to characterize the different samples within the scope of 500–4000 cm^−1^ with 32 repetitions in each scan, at a resolution of 4 cm^−1^.

Thermogravimetric analysis (TGA) was conducted on a STA449F3 thermogravimetric analyzer (Netzsch, Bavaria, Germany). TGA and derivative thermogram (DTG) data were collected ranging from room temperature to 800 °C at a heating rate of 10 °C/min under nitrogen and air environment, respectively.

The combustion conditions of control and modified lyocell samples in vertical condition were evaluated by YG815B vertical fabric FR tester (Nantong Sansi Electromechanical & Electrical Technology Co., Ltd., Jiangsu, China) according to ASTM D6413-99 standard.

The burning property of the specimens with the size of 100 mm × 100 mm × 2 mm was carried out on a cone calorimeter (FTT, West Sussex, UK) based on ISO 5660-1. The radiant heat flux of the horizontal structure was 35 kW/m^2^. The parameters, such as time to ignite (TTI), peak of HRR (PHRR), heat release rate (HRR), total heat release (THR), smoke production rate (SPR), and total smoke production (TSP) were assessed.

The XPLORA PLUS Raman spectrometer was utilized to measure the residual char ranging from 500 to 2500 cm^−1^.

Thermogravimetry-infrared (TG-IR) spectra of the control and modified lyocell samples were implemented on an STA 6000 Frontier TGA (Perkin Elmer, Waltham, MA, USA) combined with a Nicolet FTIR (Thermo Fisher Science, Waltham, MA, USA). The test was performed at a nitrogen flow rate of 50 mL/min. The generated gases were characterized by an infrared spectrometer with a resolution of 2 cm^−1^ ranging from 500 to 4000 cm^−1^.

## 3. Results and Discussion

### 3.1. Structure Characterization and Thermal Properties of Flame Retardants

The surface chemical composition of the flame retardant was investigated by XPS, as shown in Figure 3a, the strong peaks at 286 eV and 533 eV correspond to carbon (C) and oxygen (O) atoms with the content of 40.13 at% and 28.37 at%, respectively [35]. Two peaks at 400 eV and 134 eV are attributed to nitrogen (N) and phosphorus (P) components, respectively [34]. This indicated phosphorus has been successfully introduced into the expected flame retardant structure. FTIR spectrum was conducted to explore the chemical bonds of flame retardant. As shown in Figure 3b, the characteristic peak which appeared at 1283 cm^−1^ belonged to P=O stretching [36]. The peak located at 3304 cm^−1^ was attributed to O-H stretching [37]. The characteristic peaks around 1633 and 1529 cm^−1^ were assigned to C=O stretching, N-H wagging, and C-H bending, respectively [38,39]. The peak at 943 cm^−1^ corresponds to P-O-N stretching [40]. The results indicated that the predicated flame retardant was synthesized as shown in Figure 1.

### 3.2. Element Composition and Surface Morphologies of the Lyocell Fibers

#### 3.2.1. XPS Analysis

Figure 4 showed the XPS spectra of the control and modified lyocell samples. For the blank fabric, peaks at 286 eV and 533 eV corresponding to carbon and oxygen atoms were observed. Moreover, phosphorus and nitrogen were not detected because the sample was carbohydrate and mainly composed of carbon and oxygen. Compared to the control sample, two new characteristic peaks at 134 eV and 400 eV of FR-lyocell sample were observed, ascribed to P_2p_ and N_1s_, respectively [35,41]. Table 2 lists the elemental contents of the pristine and modified lyocell samples. The amount of phosphorus and nitrogen of the modified fabric was 1.2 wt% and 9.5 wt%, respectively, indicating the flame retardant was successfully introduced into the modified lyocell fabric.

#### 3.2.2. SEM and EDS Analysis

To inspect the distribution morphology of the flame retardant on the surface of the modified lyocell fabric, the mapping of phosphorus (P) and nitrogen (N) was performed by energy dispersion spectroscopy (EDS), as displayed in Figure 5. Figure 5a,b showed that the surface of the virgin fabric was glazed, while the surface of the modified fabric became rough. As shown in Figure 5d,e, it can be observed that phosphorus and nitrogen elements appear on the modified fabric, indicating that the flame retardant changes the external morphology and element percentages of the fabric. It can be seen from Figure 5c that C, N, O, and P elements are uniformly distributed on the surface of the modified lyocell fabric.

### 3.3. FTIR Analysis

Figure 6a exhibits the FTIR spectra of the original and modified lyocell fabrics. Both spectra are similar, and many characteristic absorption peaks are in the same location. The characteristic peak at 1660 cm^−1^ is assigned to the stretching vibrations of C=O group [42]. The FR-lyocell fabric showed a band at 2890 cm^−1^ on account of the C-H stretching bands [43]. Additionally, the peaks at 1022 and 896 cm^−1^ were allotted to the P=O [35,41] and P-O-C group, respectively [44,45]. Hence, the FTIR and XPS results indicate that the flame retardant has been bonded to the surface of lyocell fabric. Figure 6b presented the FTIR spectra of dicyandiamide and flame retardant. The dicyandiamide showed -NH_2_ absorption peaks at 3000–3500 cm^−1^. The peak at 2208 cm^−1^ belonged to the absorption peak of -CN. The characteristic peak at 1540 cm^−1^ was assigned to N-H bending vibration peak. As previously reported in the literature [44,46,47], dicyandiamide mainly played a catalytic role and did not participate in the reaction. Therefore, the characteristic peaks of dicyandiamide did not appear in the infrared spectrum of flame-retardant fabric.

### 3.4. Thermal Stability Analysis

Figure 7 present the thermogravimetric (TG) and derivative thermogravimetric (DTG) curves of the control and FR-lyocell under N_2_ and air flow. The key data are summed up in Table 3. In nitrogen atmosphere, the thermal degradation of the original lyocell fabric shows three weight loss stages. The first one takes place between 35 °C and 150 °C in which lyocell fabric loses bounded water as temperature increases. The weight loss of lyocell fabric decreases sharply in the second stage, which begins at 296 °C and finishes at 362 °C. In this process, the maximum degradation rate reaches 340 °C, and the mass loss is 52.1%. The third one is above 390 °C, and the char residue of lyocell fabric attributed to the previous two stages continues to decompose slowly, leaving only 13.6% char residue at 800 °C. In summary, at low temperature, the degradation of lyocell fabric is a process of gradual degradation, including depolymerization, hydrolysis, and dehydration reactions. At high temperature, the crystalline area of lyocell will decompose to generate L-glucose, and then produce combustible gases and a small amount of residual char. Compared to the control sample, the degradation process of lyocell fabric after flame retardant modification changes markedly. The modified lyocell sample has a large weight loss rate between 256 °C and 290 °C, which moves towards lower temperature signally. The modified sample reaches the maximum degradation rate at 278 °C, which shifts to a lower temperature. In addition, the char residue of modified lyocell fabric is 33.7% at 800 °C, which is obviously higher than that of the blank fabric. These characterizations manifest that the high temperature thermal stability of lyocell fabric is heightened by the flame retardant.

Furthermore, to delve more deeply into the thermal oxidation stability of the fabrics, TG tests are performed in air. Although the test conditions are transformed into the air flow, the curves of the first two instances of weight loss are rather similar. The difference is that as the temperature continues to rise from 347 °C to 496 °C, a new weight loss appears. In this stage, the char residue is further oxidized, releasing flammable and nonflammable volatiles. Regardless, the amount of the char residue of FR-lyocell is still greater than that of the control sample at 700 °C, indicating the improved thermal oxidative stability and char formation ability of lyocell fabric.

### 3.5. Burning Properties Analysis

The flammability of the blank and modified lyocell samples were estimated by vertical combustion experiments. Figure 8 showed the vertical combustion results of the control sample, modified fabric before and after washing. The combustion results of the modified sample were quite different from those of the blank sample, but little different from those of the washed sample. The control sample burned rapidly and violently in the air, and the combustion ended at 40 s, with little residual char remaining at 60 s. On the contrary, the modified sample did not burn even after exposure to flame for 60 s, showing good carbonization. The combustion of the washed sample was similar to that of the modified sample except for the wider and longer char length. In detail, the char length and the width of the modified lyocell fabric were 63 mm and 17 mm, respectively. The char length and the width of the washed specimen were 66 mm and 22 mm.

In order to better assess the flame retardant properties of modified fabrics, the LOI tests were conducted, and the detailed data are listed in Table 4. The LOI value of original fabric is 15%. In contrast, the LOI value of the FR-lyocell fabric increased to 39.5%, indicating the improved flame retardancy of the modified fabric. As a wearable fabric, its flame-retardant durability is a required index. Thus, it is necessary to test the water resistance of flame-retardant lyocell fabric. The washing durability of flame-retardant lyocell fabric was estimated by LOI changes after repeated washing. After 10 washing cycles, the LOI value of the flame-retardant fabric decreased to 37%, with a small decrease, indicating satisfactory washable performance. These phenomena demonstrated that FR-lyocell fabric possessed good flame retardancy and washing durability.

### 3.6. Cone Calorimeter Analysis

Cone calorimeter tests were carried out to imitate the true combustion condition and to assess the combustion properties of the samples. The combustion parameters, such as heat release rate (HRR), total heat release (THR), smoke production rate (SPR), and total smoke production (TSP), were analyzed. The related curves are depicted in Figure 9, and the corresponding data are listed in Table 5. As shown in Figure 9a, the HRR curve of the primordial fabric depicted a prominent peak at around 45.0 s with a PHRR close to 144.7 kW/m^2^. It demonstrated that the untreated fabric burned rapidly after exposure to the flame, giving off a lot of heat to support the fabric to continue burning [48]. After flame-retardant modification, the PHRR decreased dramatically, which was relegated to 15.4 kW/m^2^ at 25.0 s. The THR of the modified fabric diminished from 5.5 to 2.4 MJ/m^2^. The PHRR and THR values of the treated lyocell fabric in this work are lower than those of the flame-retardant lyocell fibers reported in the literature (17.9 kW/m^2^ and 5.1 MJ/m^2^, respectively) [49]. Moreover, the THR value of the work is lower than that of the research reported by Luo (3.2 MJ/m^2^) [41].

The fire growth rate index (FIGRA) represents the fire scale and the flame growth rate, and it is calculated in proportion to the PHRR value and the time needed to achieve the value. The FIGRA value of the treated sample dwindled from 3.2 to 0.6 kW/m^2^s. The char yield increased from 8.8 wt% to 34.3 wt% due to the flame-retardant treatment, as evidenced in Table 4. The conical calorimetry images of the FR-lyocell sample before and after combustion are shown in Figure 10. The char residue of the FR-lyocell fabric was dense and relatively complete. These results suggested that the modified lyocell fabric possessed superior flame retardancy and char-forming ability.

Smoke is one of the important causes of death in fire [50]. Flammable gases intensify the combustion while incombustible gases weaken it. Therefore, the smoke production rate (SPR) and total smoke production (TSP) are the crucial indicators to appraise the flammability properties of the samples. Obviously, the TSP value of the modified specimen was much higher than that of the pristine fabric, which might be due to the fact that flame-retardant modification restrained the complete combustion of the fabric and a large amount of smoke was produced. At the same time, the modified sample showed many new peaks in SPR which were not found in the original sample. Thus, the flame retardant changed the gases generation mechanism during the pyrolysis process of the lyocell fabric.

### 3.7. Roman Spectrum

Roman spectrum was performed to assay the char residue structure of the finishing fabric, which helped to better comprehend the flame-retardant mechanism. As shown in Figure 11, the strong absorption bands of the modified lyocell fabric appeared at 1578 cm^−1^ and 1368 cm^−1^, which were G-band of graphite structure carbon and D-band of amorphous structure carbon, severally [51]. The *I*_D_/*I*_G_ ratio (where *I*_D_ and *I*_G_ are the peak intensities of D-band and G-band, respectively) is generally applied to appraise the quality of carbon substances. The *I*_D_/*I*_G_ value of the char residue of FR-lyocell is 0.867, which is at a lower level, and indicates that a highly graphitized char layer is produced during the burning process of the FR-lyocell fabric. Accordingly, both graphite and amorphous carbon are contained in the residue of the modified specimen. In contrast, the control sample burns violently with little char residual, and cannot be availably evaluated by Raman spectroscopy. The results suggest that flame retardant treatment can effectually promote the carbonization of the lyocell fabric during the combustion process, as evidenced by cone calorimeter test, which is consistent with the results of TG analysis. The increased char yield is expected to inhibit the decomposition of the lyocell fabric. Moreover, the char generated on the lyocell fabric during the combustion process can prevent external oxygen from entering the combustion zone and at the same act as a heat barrier. The results demonstrated that the flame retardant modification is beneficial to the generation of char residue and the production of incombustible substances during the burning process, which conforms to the condensed phase flame-retardant mechanism.

### 3.8. TG-IR Analysis

The pyrolysis products of the blank and finished lyocell samples were investigated by TG-IR technique. The TG-IR spectra and their related 3D spectra are shown in Figure 12 and Figure 13, respectively. The gases produced of the pristine fabric between 100 and 650 °C were analyzed. For the control fabric, the peak at 3594 cm^−1^ was ascribed to the stretching vibration of -OH group of the steam [52]. The sharp peak at 1744 cm^−1^ was the result of C=O vibration from carbonyl compounds [53]. The strong absorption peak at 1065 cm^−1^ was due to the C-O-C bond of the fabric and the obvious peak at 2362 cm^−1^ was attributed to the shrinkage vibration peak of CO_2_ [41,48]. The peaks at 2914 cm^−1^ and 2974 cm^−1^ arisen at 340 °C correspond to the vibration absorption of C-H group [54]. It is clear that the most intense pyrolysis stage occurred at 340 °C, which can be identified with the TG results. At 300–340 °C, the intensity of -OH vibration increased gradually as the temperature rose, and reached a maximum at 340 °C, then the peak decreased little by little, but did not disappear completely at 650 °C. The release tendency of CO_2_ was similar to that of -OH. In summary, the major degradation products of virgin lyocell fabric comprise flammable (such as hydrocarbons) and non-flammable (such as CO_2_ and H_2_O) substances.

For the treated lyocell fabric, the peak intensity of the modified specimen at 1744 cm^−1^ and 1065 cm^−1^ was significantly lower than that of the original fabric. A new absorption peak emerged at 1527 cm^−1^, attributed to the stretching vibration of C-N and bending vibration of NH_3_, NO_2_ [55]. The absorption peaks at 2914 cm^−1^ and 2974 cm^−1^ disappeared, indicating that the flame retardant inhibited the formation of flammable alkanes. The appearance of new peaks and the disappearance of old peaks confirmed that the generation of combustible gases was significantly reduced and incombustible gas (such as CO_2_, NH_3_, NO_2_) were produced during the thermal decomposition of the modified sample.

The temperature of CO_2_ released from the modified sample (100 °C) was lower than that of the blank sample (220 °C), which was considered to be the decomposition of the flame retardant. In general, the trend of peak intensity of CO_2_ increased as the temperature rose from 100 °C to 800 °C. Moreover, the temperature corresponding to the generation of H_2_O shifted to a lower temperature. These results are consistent with the conical calorimetry results, indicating that incombustible substances (H_2_O and CO_2_) are generated which not only prevent the exterior oxygen from entering the combustion region but also dilute the combustible gases in the atmosphere [27]. Therefore, the combustion is availably restrained and the flame retardancy of the lyocell fabric is effectively improved. Based on the above analysis, the flame retardant mechanism of FR-lyocell fabrics was proposed in Figure 14.

## 4. Conclusions

A novel, high-efficiency, halogen-free, and formaldehyde-free flame retardant containing phosphorus and nitrogen was successfully synthesized by phosphorylation of arginine. Afterwards, the flame retardant was grafted onto the lyocell fabric to prepare washable and durable flame-retardant lyocell fabrics. XPS and Fourier-transform infrared spectrum analysis indicated that the flame retardant containing phosphorus and nitrogen elements covalently bonded with the lyocell fabric. TG analysis in N_2_ indicated that the temperature required for the maximum degradation rate of the modified lyocell fabric decreased by almost 60 °C, and the char residue in nitrogen increased from 13.7% to 33.3%. The vertical combustion test proved that the modified and washed lyocell fabrics possessed excellent flame retardancy and composed a complete and dense char layer after combustion. The lower PHRR and THR values of the modified specimens were found in the cone calorimetry test, indicating excellent heat insulation property and flame retardancy. TG-IR and Raman spectroscopy analyses indicated that the flame-retardant mechanism of the modified fabric mainly followed the flame-retardant mechanism of condensed phase and gas phase.

## Figures and Tables

**Figure 1 molecules-26-03588-f001:**
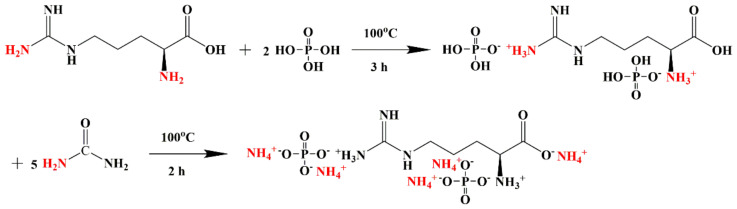
Synthesis principle of flame retardant.

**Figure 2 molecules-26-03588-f002:**
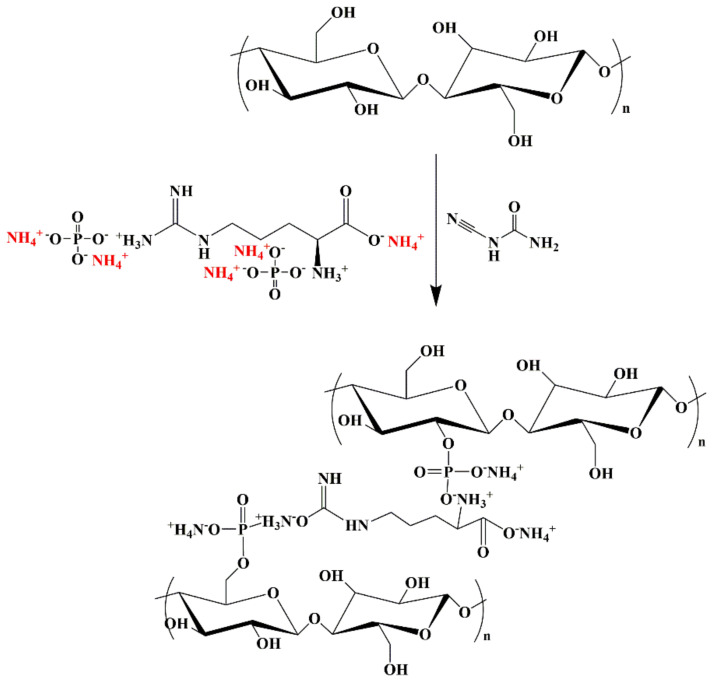
Synthetic route of flame−retardant lyocell fabric.

**Figure 3 molecules-26-03588-f003:**
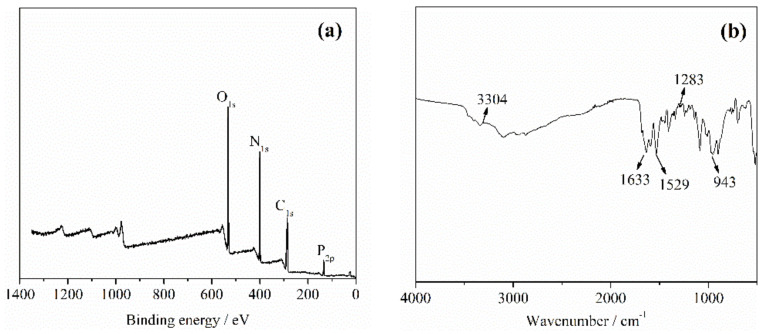
XPS (**a**) and FTIR (**b**) spectra of flame retardant.

**Figure 4 molecules-26-03588-f004:**
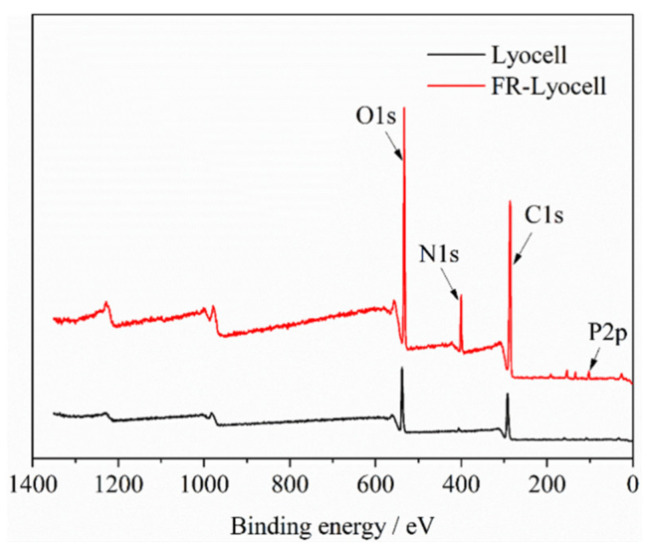
XPS spectra of lyocell and FR−lyocell samples.

**Figure 5 molecules-26-03588-f005:**
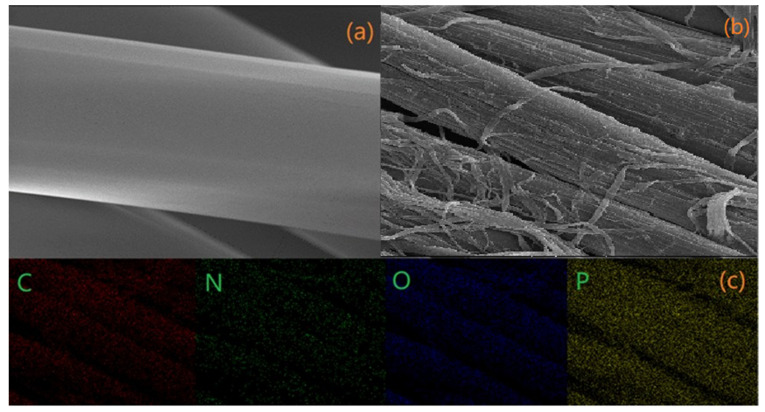
SEM images of lyocell (**a**) and FR−lyocell (**b**) fabrics; SEM−mapping images of FR−lyocell (**c**), EDX spectra of FR−lyocell (**d**), and lyocell (**e**) fabrics, respectively.

**Figure 6 molecules-26-03588-f006:**
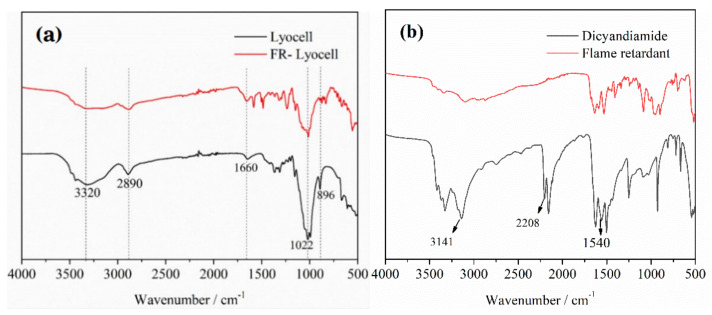
FTIR spectra of lyocell and FR−lyocell samples (**a**), Dicyandiamide and Flame retardant (**b**).

**Figure 7 molecules-26-03588-f007:**
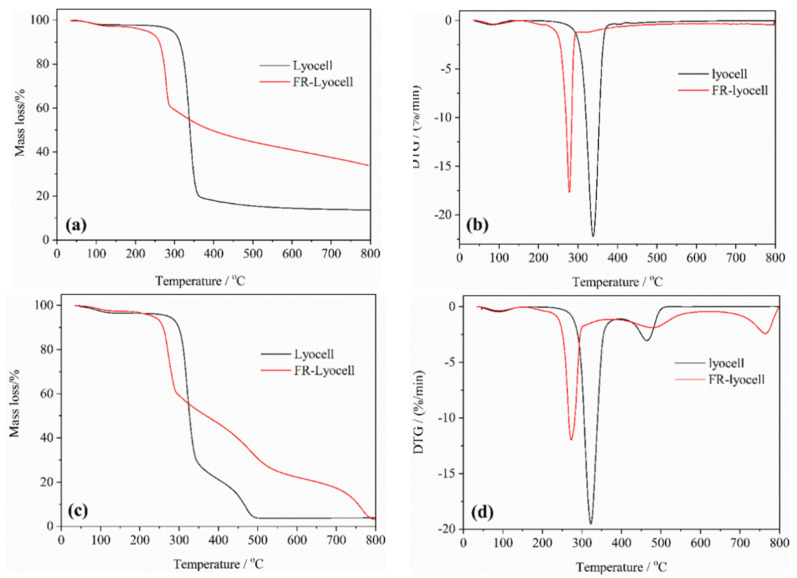
TG curves of lyocell and FR−lyocell fabrics in N_2_ (**a**); DTG curves of lyocell and FR−lyocell fabrics in N_2_ (**b**); TG curves of lyocell and FR−lyocell fabrics in Air (**c**); DTG curves of lyocell and FR−lyocell fabrics in Air (**d**).

**Figure 8 molecules-26-03588-f008:**
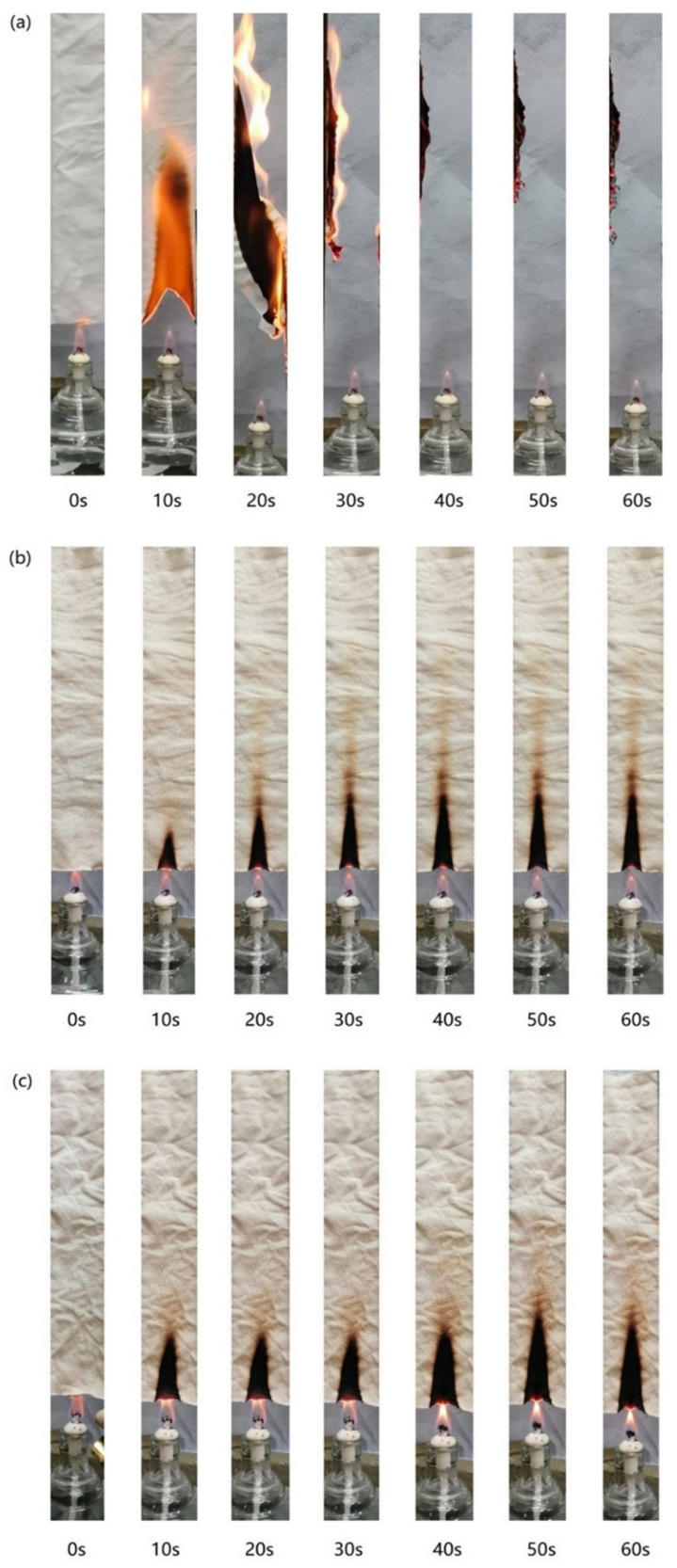
Vertical combustion test results of control (**a**), FR−lyocell before (**b**) and after washing (**c**).

**Figure 9 molecules-26-03588-f009:**
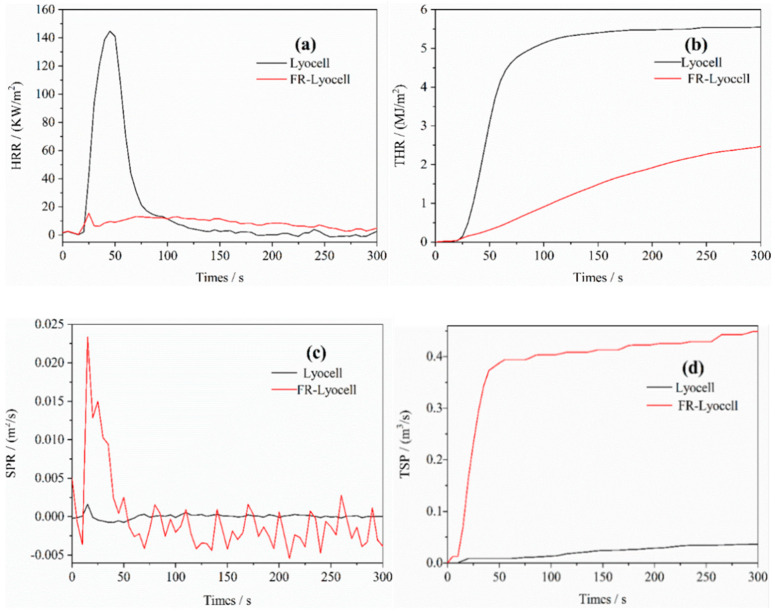
HRR (**a**), THR (**b**), SPR (**c**), and TSP (**d**) curves of lyocell and FR−lyocell fabrics.

**Figure 10 molecules-26-03588-f010:**
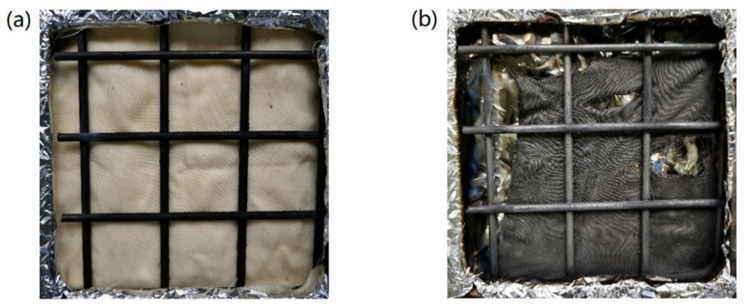
Conical calorimetry images of FR−lyocell before (**a**) and after (**b**) combustion.

**Figure 11 molecules-26-03588-f011:**
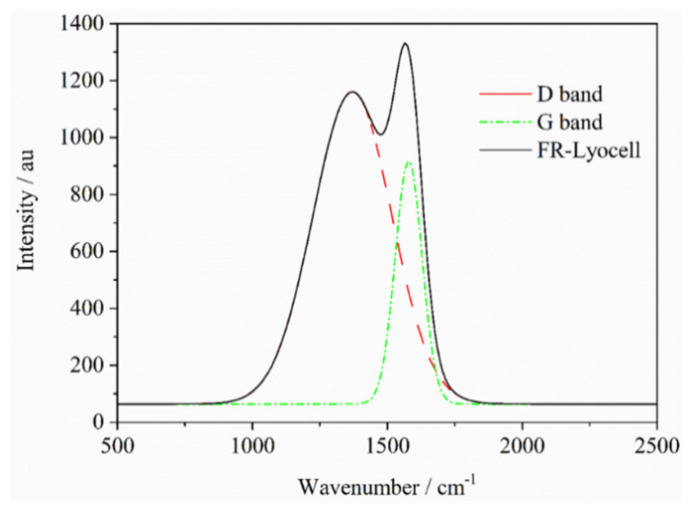
Raman spectra of the char residues of FR−lyocell fabrics.

**Figure 12 molecules-26-03588-f012:**
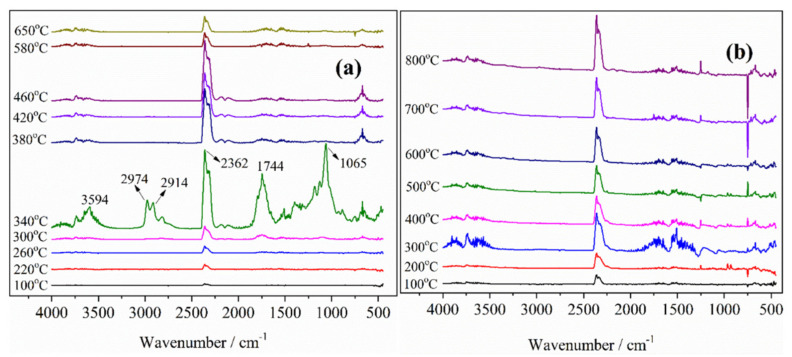
TG−IR spectra of lyocell fabric (**a**) and FR−lyocell fabrics (**b**).

**Figure 13 molecules-26-03588-f013:**
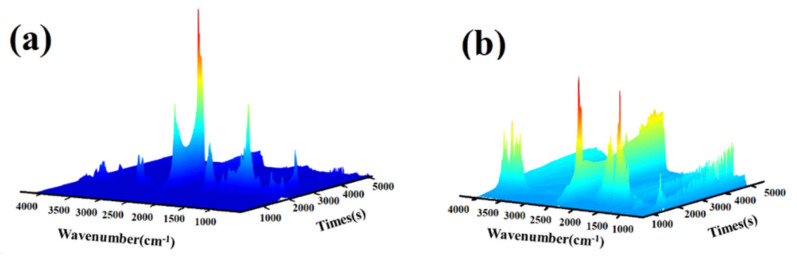
TG−IR 3D images of gaseous compounds during thermal degradation process for control (**a**) and treated (**b**) lyocell fabrics.

**Figure 14 molecules-26-03588-f014:**
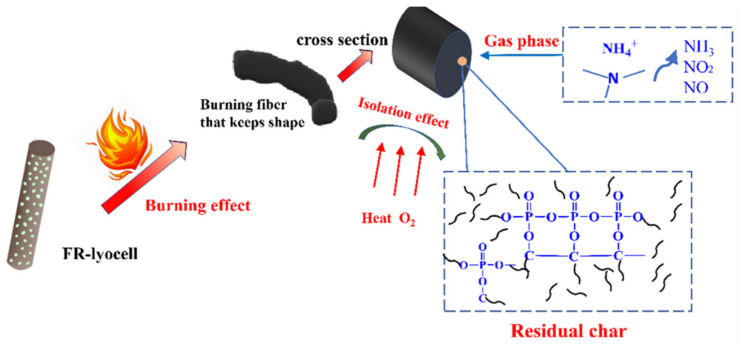
Schematic illustration mechanism of fire resistant.

**Table 1 molecules-26-03588-t001:** Weight gain (WG) of lyocell fabrics treated with 120 g/L, 150 g/L, and 180 g/L finishing solutions.

Sample	Concentration (g/L)	Weight Gain (%)	Average Weight Gain (%)
1-1	120	5.3	7.3
1-2	120	6.7	
1-3	120	10.0	
2-1	150	7.3	8.9
2-2	150	8.7	
2-3	150	10.7	
3-1	180	11.3	11.5
3-2	180	10.0	
3-3	180	13.1	

**Table 2 molecules-26-03588-t002:** Element composition of lyocell and FR−lyocell samples.

Sample	C (wt%)	O (wt%)	N (wt%)	P (wt%)
Lyocell	64.5	30.6	-	-
FR-lyocell	61.2	25.7	9.5	1.2

**Table 3 molecules-26-03588-t003:** TG analysis data of lyocell and FR−lyocell fabrics in N_2_ and air atmosphere.

Atmosphere	Sample	T_5%_ (°C)	T_50%_ (°C)	T _max_ (°C)	Residue at 800 °C (wt%)
N_2_	lyocell	295.0	339.5	337.3	13.6
	FR-lyocell	229.4	394.1	278.1	33.7
Air	lyocell	273.0	326.5	333.0	3.9
	FR-lyocell	234.1	369.1	272.8	3.3

**Table 4 molecules-26-03588-t004:** LOI values of the samples with different washing cycles.

Sample	LOI Values (%)					
	0	2	4	6	8	10
lyocell	17.0					
FR-lyocell	39.5 ± 0.1	39.0 ± 0.1	38.8 ± 0.1	38.5 ± 0.1	38.5 ± 0.1	37 ± 0.1

**Table 5 molecules-26-03588-t005:** Parameters of lyocell and FR−lyocell fabrics acquired from CC test.

Sample	TTI (s)	PHRR (kW/m^2^)	T_PHRR_ (s)	THR (MJ/m^2^)	PSPR (m^2^/s)	TSP (m^2^)	Residue (wt%)	FIGRA (kW/m^2^s)
Lyocell	18.0	144.7	45.0	5.5	0	0	8.8	3.2
FR-lyocell	16.0	15.4	25.0	2.4	-	0.4	34.3	0.6

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
