# Peer review of "Synthesis of Novel Arginine-Based Flame Retardant and Its Application in Lyocell Fabric"

_molecules, 2021, doi:10.3390/molecules26123588_

Round 1

Reviewer 1 Report

The manuscript by J. Chen, Y. Liu, J. Zhang, Y. Ren and X. Liu entitled ”Synthesis of novel arginine-based flame retardant and its application in lyocell fabric” concerns the production of flame-retardant lyocell fabrics with the use of biomass-based flame retardant. However, the main problem of the reviewed manuscript is the lack of novelty. The review of literature has been made in a piecemeal way. The authors presented a very superficial and selective literature review, which does not reflect the actual picture of this research area. Research on the fire resistance of lyocell fibers has been carried out for many years (Polym. Degrad. Stab. 54, 1996, 401; Polym. Degrad. Stab. 64, 1999, 505). Moreover, in the last few years many works have been published (also by the authors of this paper) in this subject area. Here are some examples: Carbon Lett. 17(1), 2016, 74; Cellulose 25, 2018, 6745; Carbohydrate Polym. 153, 2016, 78; Cellulose Chem. Technol. 54(9-10), 2020, 1015. The discussion of these works in the Introduction would allow the readers to define the novelty of the research carried out and compare the effectiveness of the arginine-based flame retardant. On the other hand, the current publication and presentation of the results is very similar to the previous work of the authors (Cellulose 27, 2020, 6677), in which they used a vitamin C-based modifier instead of arginine. The authors also do not mention a large and important group of flame retardants based on silicon compounds. In addition, there is a lot of ambiguity in the experimental part. For example, in the preparation of a flame retardant compound, the amount of arginine that was used to prepare the solution was given, but this amount is twice as high as the solubility of arginine in water. Furthermore, if the arginine flame retardant is a new compound, a full characterization of this compound should be provided. How was the synthesis going? What was the efficiency of this reaction? The authors gave a vague description of the flame retardant compound content in the modifying composition. First, they state that 120g / L, 150g / L or 180g / L were used (these values ​​should be expressed as a percentage) and then they state that a 1:50 bath was used. What is the ratio? Although the experimental part describes the preparation of three compositions (with different concentrations), all results apply to only one composition (with the highest concentration). What about the rest of the compositions? Most of the flammability test results presented in the article concern unwashed samples. LOI analysis should be performed for all modified samples. These results should be compared with the results for the samples after repeated washing. The analysis of SEM images also does not contribute to the elucidation of results because no clear differences are visible. Where do the Au and Pt signals on the EDX spectra come from?

The authors have proposed a crosslinking mechanism (presented in Fig. 2) that is not supported by any evidence. Moreover, the cross-linking agent was reported to be dicyandiamide but there is another compound in the figure. The FT-IR spectrum of the pure modifier should be made and compared with the spectrum of the modified and pure fabric. The presented spectra are very similar (as well as other research results) to those presented in Cellulose 27, 2020, 6677. Figures 11 and 12 do not exactly match each other. In addition, the authors propose a fire protection mechanism, evidence for which was not supported by the gas analysis.

The authors present all the obtained results but do not comment on them. Above all, they do not compare these results to the results of other studies, and such a comparison would allow for the identification of novelty and innovative elements of the developed modifier.

Summing up, the manuscript requires a thorough reedition and addition of the missing data. Moreover, I do not see the novelty in the submitted work; therefore, I do not recommend its publication in „Molecules”.

Author Response

Dear reviewer

Thank you for your questions and suggestions. Our reply is attached."Please see the attachment.

Reviewer 2 Report

The submitted article describes the preparation of washable and durable flame retardant lyocell fabrics by the grafting of high-efficiency flame retardant containing phosphorus and nitrogen. The authors proposed to obtain halogen- and formaldehyde-free flame retardant by the phosphorylation of arginine. Design of modified lyocell fabric was successfully confirmed by a number of physical methods. The manuscript is well-written and properly organized.

Here are some questions and suggestions for improvements.

  • Why did the authors used urea for the synthesis of flame retardant? Could ammonia, for example, be used for this synthesis?
  • Figures 1 and 2: there is some mistake in the image of the structure of flame retardant. I think there is extra oxygen atom in the butylene fragment.
  • –line 361: reference 4, “4Li, P.; Wang, B.;…” could be changed to “Li, P.; Wang, B.;…”

Author Response

Dear reviewer

Thank you for your questions and suggestions. Our reply is attached. Please see the attachment

Round 2

Reviewer 1 Report

The manuscript’s revision took into account most of my comments. It may be published in its current form.